# Towards Unsupervised Classification with Deep Generative Models

## Abstract

Deep generative models have advanced the state-of-the-art in semi-supervised classification, however their capacity for deriving useful discriminative features in a completely unsupervised fashion for classification in difficult real-world data sets, where adequate manifold separation is required has not been adequately explored. Most methods rely on defining a pipeline of deriving features via generative modeling and then applying clustering algorithms, separating the modeling and discriminative processes. We propose a deep hierarchical generative model which uses a mixture of discrete and continuous distributions to learn to effectively separate the different data manifolds and is trainable end-to-end. We show that by specifying the form of the discrete variable distribution we are imposing a specific structure on the model's latent representations. We test our model's discriminative performance on the task of chronic lymphocytic leukemia (CLL) diagnosis against baselines from the field of computational flow cytometry (FC), as well as the Variational Autoencoder literature.

## 1 Introduction

Variational Autoencoders (VAEs) have recently shown remarkable performance in unsupervised generative modeling of high-dimensional data generated by complex distributions (Kingma & Welling, 2013; Rezende et al., 2014), as well as semi-supervised classification where only a small subset of the data set is labeled (Kingma et al., 2014). While the interaction between the generative and the classification capabilities of semi-supervised models has been recently explored in literature (Maaløe et al., 2016; 2017), there has been little investigation of the discriminative capabilities of a purely unsupervised framework with most works focusing on the task of unsupervised clustering (Johnson et al., 2016; Xie et al., 2016; Jiang et al., 2017). Furthermore, most of these works have been evaluated on mostly benchmark data sets which do not capture the difficulties that are often encountered on real-world data. For instance, there has been no investigation of the performance of these methods on data sets with significant class imbalance. The question that is, then, posed is whether deep generative models can be used effectively as unsupervised classifiers, which can, in essence, be cast into a question of what type of features and architectural choices are required to achieve good classification performance in an unsupervised manner.

To examine the aforementioned questions we propose a deep hierarchical generative model and evaluate its performance on a difficult real world data set. In principle, we train our model in a completely unsupervised fashion, however in our experiments we rely on labeled data to measure our model's performance using suitable metrics for the problem domain, as well as derive a stopping criterion for training. Our model outperforms established state-of-the-art baselines used in the field of the problem domain. Our contributions are summarized in the following:

- A framework which utilizes a hierarchy of continuous representations which conclude in a discrete variable explicitly representing categories, resulting in complex, expressive, invariant and interpretable representations (Bengio et al., 2013), which are crucial in separating widely overlapping manifolds and achieve good classification results in significantly imbalanced data sets.

- Controllable representation structure through specification of the form of the aforementioned discrete variable which better suits the task at hand given a problem scenario.

## 2   TECHNICAL BACKGROUND

In this section we review techniques and frameworks that provide a useful backdrop for the derivation of our own model.

### 2.1   DEEP GENERATIVE MODELS

Variational Autoencoders (VAEs) as presented in Kingma & Welling (2013) use at most two layers of stochastic latent variables, however generalizations to multiple layers of latent variables have since been introduced (Rezende et al., 2014; Burda et al., 2015; Sønderby et al., 2016).

Generation in a deep generative model is achieved by a top-down stochastic pass through the model defined as:

$$p_\theta(\mathbf{x}) = p_\theta(\mathbf{z^L})p_\theta(\mathbf{z^{L-1}}|\mathbf{z^L})...p_\theta(\mathbf{x}|\mathbf{z^1}) \tag{1}$$

where $L$ is the number of stochastic hidden layers and $\mathbf{z}^L$ denotes the number of latent variables in the layer. As in a standard VAE the dependence of each layer on the previous one is considered to be nonlinear and is modeled by multi-layered perceptrons (MLPs). Similarly, inference is carried out by a bottom-up stochastic pass through the model's layers:

$$q_\phi(\mathbf{z}|\mathbf{x}) = q_\phi(\mathbf{z^1}|\mathbf{x})q_\phi(\mathbf{z^2}|\mathbf{z^1})...q_\phi(\mathbf{z^L}|\mathbf{z^{L-1}}) \tag{2}$$

Optimization of a deep generative model is akin to that of a standard VAE. Namely, the reparameterization trick (Kingma & Welling, 2013) is being applied to each layer of stochastic latent variables.

### 2.2   CONTINUOUS RELAXATIONS OF DISCRETE RANDOM VARIABLES

Until recently, models trained with the variational objective have been employing mainly Gaussian latent stochastic variables, optimizing indirectly through discrete variables wherever they were used (e.g. Kingma et al. (2014) integrate over the discrete variable). This was due to the inability of backpropagating through discrete variables because of their discontinuous nature.

Jang et al. (2016) and Maddison et al. (2016) independently developed a continuous relaxation of discrete random variables. The resulting distribution was presented as the *Gumbel-Softmax distribution* and the *Concrete distribution* respectively but essentially has the same functional form. From this point on, for the sake of clarity we will adopt the latter name to refer to this distribution.

A simple way to sample from a discrete distribution is to employ the Gumbel-Max trick (Gumbel, 1954; Papandreou & Yuille, 2011; Maddison et al., 2014). The Gumbel distribution produces samples according to $-\log(-\log(u))$ where $u \sim \text{Uniform}(0,1)$. Given parameters $\alpha_1, ..., \alpha_k$ and samples $g_i$ from Gumbel(0,1), samples $z$ from the Categorical distribution can be drawn according to:

$$z = \arg\max_i(g_i + \log\alpha_i) \tag{3}$$

where z is represented as a one-hot vector. Samples from the Concrete distribution are produced by replacing the argmax operation with the softmax function:

$$y_i = \frac{\exp((\log\alpha_i + g_i)/\tau)}{\sum_{j=1}^{k}\exp((\log\alpha_j + g_j)/\tau)} \qquad \text{for i = 1, ..., k}$$

Crucially, the reparameterization trick can now be applied in a similar manner to Gaussian samples. The probability density function of the Concrete distribution is the following:

$$p_{\alpha,\tau}(y_1, ...y_k) = \Gamma(k)\tau^{k-1} \left( \sum_{i=1}^{k} \alpha_i / y_i^{\tau} \right)^{-k} \prod_{i=1}^{k} (\alpha_i / y_i^{\tau+1}) \tag{4}$$

As the temperature parameter $\tau$ approaches 0, samples from the Concrete distribution more accurately approximate one-hot samples from a Categorical distribution and in the limit, the two distributions become identical. We refer the reader to Jang et al. (2016); Maddison et al. (2016) for more details.

## 3 METHOD

In this section we present the problem setting and our proposed model. Additionally, we motivate our model design choices and modifications to the variational objective.

### 3.1 PROBLEM

Chronic lymphocytic leukemia (CLL) is the most common form of leukemia in adults in Western countries (Hallek, 2013). CLL is usually diagnosed during routine blood tests and Flow Cytometry (FC) is one of the examination procedures used for confirming the diagnosis (Hallek, 2013). Flow cytometry (FC) is a powerful technique for single cell analysis, which is routinely used for the diagnosis of haematological malignancies (Brown & Wittwer, 2000), however it is heavily dependent on the experience of the expert performing it, oftentimes resulting in serious discrepancies across experts.

During or after treatment of CLL, the term Minimal Residual Disease (MRD) is used to define the small amount of leukemic cells detected in patients (typically 10s to a few 100s of leukemic cells in samples of 500,000 cells or more). Particularly for CLL, the limit for MRD positive diagnosis is set at 1 leukemic cell per 10,000 white blood cells in blood or bone marrow (Böttcher et al., 2013). The problem, from a manifold learning and generative modeling point of view is to adequately separate the two data manifolds of healthy and leukemic cells.

This problem is difficult not just because of the sheer size of the healthy cell population, which leads to significant manifold overlap, but also because there are other manifolds present in the data, e.g. representing different types of cell populations in the blood sample, with many different factors of variation, some of which are known (cell granulation, cell size, etc.) and some of which may be unknown. Most clustering algorithms, which are traditionally used in computational FC, particularly those that "infer" the number of clusters automatically, are unable to separate the manifolds of interest given a particular problem, because they act directly on the input/data space without making any assumptions about the latent structure present in a data set. As a result they are sensitive to noise and ultimately one has to resort to merging clusters. Furthermore, for the clustering results to be interpretable, significant amounts of hyperparameter tuning are necessary during the clustering proper, the cluster merging phase or both (e.g. number of nearest neighbors to be taken into account, appropriate distance metric, linkage, etc) resulting in an impractical, computationally expensive overall solution.

### 3.2 DEEP GENERATIVE MODELS AS FEATURE REPRESENTATION LEARNERS

An alternative to clustering algorithms would be to learn a low-dimensional feature mapping as in traditional deterministic autoencoders and then perform clustering in feature space, optimizing the two tasks either separately or jointly (Xie et al., 2016). While this is a viable strategy, possibly alleviating some of the problems discussed above, depending on the problem domain, the deterministic mapping could potentially lack expressiveness due to the low-dimensionality requirement, as it would need to compress information to avoid the curse of dimensionality. Furthermore, these methods are sensitive to noise and so, they overfit the data, being unable to disentangle the different factors of variation (Bengio et al., 2013).

A practical and principled solution in this problem domain should be able to adequately separate the multiple manifolds underlying the data and model the signal of interest, remaining invariant to

factors of variation. Furthermore, the candidate model should be able to learn feature representations that enforce a structure that facilitates the above and as a result be able to correctly classify each cell in an unsupervised fashion. Ideally it should be trained and optimized end-to-end without resorting to clustering algorithms that are separated from the feature representation learning process.

Latent variable models like VAEs and deep generative models seem like an ideal fit to the above description. They address the problem of generation of new samples with the help of a *recognition/inference model* which learns the global structure of the training data. In theory, by employing this encoder, such a model could learn representations that would prove useful in unsupervised discriminative tasks. In practice, however, this process is a lot more complicated. First of all, recognition models with inadequate expressiveness are known to encode only local information, because they model well only the regions of the posterior distribution that lie near the training examples, being unable to generalize to the full posterior manifold (Yeung et al., 2017). When one uses powerful and expressive generative models as in Bowman et al. (2015); Serban et al. (2017); Fraccaro et al. (2016), this is further exacerbated to the point where generation on the one hand and representation learning on the other, become competing tasks with the model preferring to encode mainly local information using the generative/decoding model $p(\mathbf{x}|\mathbf{z})$ and ignore the latent code (Chen et al., 2016). From a representation learning standpoint which is what we're interested in for this particular problem, the generative model is now perceived as the regularizer and we need to make sure that the recognition model is expressive enough to be able to model our true posterior as closely as possible, as well as provide an interpretable way of making a diagnosis. As such we propose a framework that addresses both these needs jointly.

## 3.3 MODEL

To address the issues discussed in 3.2 we introduce a deep generative model composed of $L$ layers of continuous stochastic latent variables. Beyond alleviating the aforementioned issues, we also aim to model a set of latent factors affecting the cell measurements and eventually, the diagnosis. We introduce a hierarchy of continuous latent variables to express these latent factors. The diagnosis is itself represented as a single (relaxed) binary variable after this cascade of layers. For the sake of brevity we will refrain from explicitly stating the continuous relaxation on this discrete variable for the remainder of this text, except when necessary. To highlight the close relationship between the discrete diagnosis and the continuous latent factors and for practical reasons we denote the mixture of the discrete and continuous variables by $\mathbf{h}$. The generative model assumes the following form:

$$p_\theta(\mathbf{h}) = p(\mathbf{y})p_\theta(\mathbf{z}|\mathbf{y}) \tag{5}$$

with

$$p_\theta(\mathbf{z}|\mathbf{y}) = p_\theta(\mathbf{z_L}|\mathbf{y})...p_\theta(\mathbf{z_1}|\mathbf{z_2}) \tag{6}$$

where

$$\mathbf{z_k} \sim N(\mu_\theta(\mathbf{z_{k+1}}), \Sigma_\theta(\mathbf{z_{k+1}})) \quad \text{for } k = 1...L-1,$$
$$\mathbf{z_L} \sim N(\mu_\theta(\mathbf{y}), \Sigma_\theta(\mathbf{y}))$$

Parameters $\mu_\theta(\cdot), \Sigma_\theta(\cdot)$ are non-linear functions of the next layer in the hierarchy computed by feedforward neural networks and $\theta$ are the generative parameters. The prior on the discrete variable $p(\mathbf{y})$ is set to be the discrete uniform probability. The observation model is defined as:

$$p_\theta(\mathbf{x}|\mathbf{h}) = p_\theta(\mathbf{x}|\mathbf{z_1}) \tag{7}$$

and describes the observations conditioned on the latent variables. The inference model is described below:

$$q_\phi(\mathbf{h}|\mathbf{x}) = q_\phi(\mathbf{z}|\mathbf{x})q_\phi(\mathbf{y}|\mathbf{z}) \tag{8}$$

where

$$q_\phi(\mathbf{z}|\mathbf{x}) = q_\phi(\mathbf{z_1}|\mathbf{x})...q_\phi(\mathbf{z_L}|\mathbf{z_{L-1}}) \tag{9}$$
$$q_\phi(\mathbf{y}|\mathbf{z}) = q_\phi(\mathbf{y}|\mathbf{z_L}) \tag{10}$$

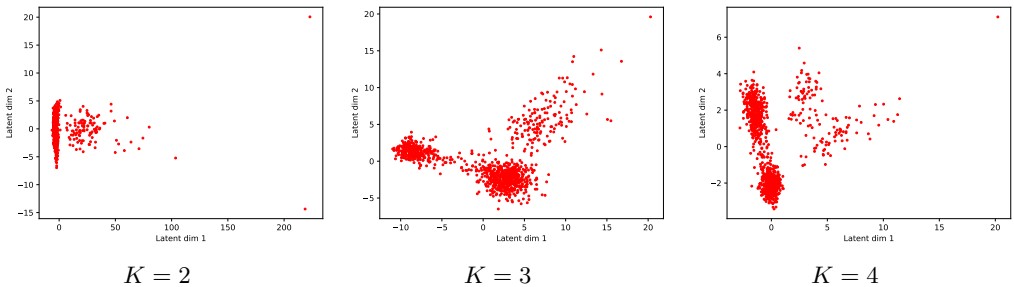

Figure 1: PCA plots of latent representations for different number of classes $K$ of discrete variable $\mathbf{y}$. Increasing $K$, imposes a different structure on the model, encouraging it to learn and separate different manifolds. E.g. setting $K = 4$ the model successfully separates the 4 different cell populations present in the data (lymphocytes, monocytes, granulocytes and destroyed red blood cells).

with

$$\mathbf{z_k} \sim N(\mu_\phi(\mathbf{z_{k-1}}), \Sigma_\phi(\mathbf{z_{k-1}})) \quad \text{for } k = 1...L,$$
$$\mathbf{y} \sim Concrete(\alpha_\phi(\mathbf{z_L}), \tau)$$

where similar to the generative model, parameters $\mu_\phi(\mathbf{z_{k-1}}), \Sigma_\phi(\mathbf{z_{k-1}}), \alpha_\phi(z_L)$ are functions that describe non-linear relationships between stochastic layers and are computed by feedforward neural networks with $\phi$ being the variational parameters. The temperature parameter $\tau$ governs the degree of relaxation of the discrete variable and can either be constant, annealed during training or learned. In the case of the first stochastic layer $\mathbf{z_1}$, layer $\mathbf{z_0}$ refers to the observed data $\mathbf{x}$. The function $\alpha_\phi(\mathbf{z_L})$ models the relationship between the last Gaussian layer and the discrete variable $\mathbf{y}$.

By conditioning on the discrete variable $\mathbf{y}$, we are enforcing a tight coupling between the latent factors of variation (represented by $\mathbf{z}$) and categorical features (represented by $\mathbf{y}$). As a result the activity of the discrete variable $\mathbf{y}$ and continuous variables $\mathbf{z_1}...\mathbf{z_L}$ is highly correlated. More than simply capturing the modes of continuous variable $\mathbf{z_L}$, the discrete variable $\mathbf{y}$ imposes a particular structure that is propagated through all the continuous layers $\mathbf{z_1}...\mathbf{z_L}$ of the feature hierarchy. This allows us a degree of control with respect to the structure of the latent variables, which we can exploit depending on the task at hand. Figure 1 illustrates such an example. By increasing $K$ we facilitate multiple manifold learning.

Both in the inference and the generative models, covariances assume a diagonal structure. Our model's overall architecture can be seen in Figure 2.

Employing a relaxed discrete variable allows us to avoid marginalization over all its possible values. At the same time, however, we are using a continuous *surrogate loss* in place of the original discrete one (Schulman et al., 2015; Maddison et al., 2016).

$$\begin{aligned}
\mathcal{L}(\mathbf{x}; \phi, \theta) &= \mathbb{E}_{q_\phi(\mathbf{h}|\mathbf{x})} \left[ \log p_\theta(\mathbf{x}, \mathbf{h}) - \log q_\phi(\mathbf{h}|\mathbf{x}) \right] \\
&= \mathbb{E}_{q_\phi(\mathbf{h}|\mathbf{x})} \left[ \log p_\theta(\mathbf{x}|\mathbf{h}) + \log p(\mathbf{h}) - \log q_\phi(\mathbf{h}|\mathbf{x}) \right] \\
&= \mathbb{E}_{q_\phi(\mathbf{h}|\mathbf{x})} \left[ \log p_\theta(\mathbf{x}|\mathbf{h}) \right] - \mathbf{KL}[q(\mathbf{h}|\mathbf{x})||p_\theta(\mathbf{h})]
\end{aligned} \tag{11}$$

To induce more distributed and thus expressive latent codes we adopt *deterministic warm-up* per Sønderby et al. (2016), where we begin training our model with a deterministic autoencoder and then gradually introduce the KL term, to prevent the approximate posterior from collapsing onto the prior early in training and disconnecting latent units. Thus we introduce a $\lambda$ term in expression 11, which we linearly anneal during training from 0 to 1:

$$\mathcal{L}(\mathbf{x}; \phi, \theta) = \mathbb{E}_{q_\phi(\mathbf{h}|\mathbf{x})} \left[ \log p_\theta(\mathbf{x}|\mathbf{h}) \right] - \lambda \, \mathbf{KL}[q(\mathbf{h}|\mathbf{x})||p_\theta(\mathbf{h})] \tag{12}$$



Figure 2: Model architecture. Left: inference model, right: generative model.

The gradient estimations of the lower bound will be biased with respect to the original discrete loss, but unbiased and low-variance with respect to the surrogate loss (Maddison et al., 2016).

To obtain low variance updates to our parameters through stochastic gradient optimization of the bound, we use the reparameterization trick for both discrete and continuous variables concurrently:

$$\nabla_{\phi,\theta}\mathcal{L}_k(x) = \nabla_{\phi,\theta}\mathbb{E}_{q_\phi(\mathbf{h}|\mathbf{x})}\left[\log p_\theta(\mathbf{x},\mathbf{h}) - \log q_\phi(\mathbf{h}|\mathbf{x})\right]$$
$$= \nabla_{\phi,\theta}\mathbb{E}_{\epsilon,g}\left[\log p_\theta(\mathbf{x},\tilde{\mathbf{h}}) - \log q_\phi(\tilde{\mathbf{h}}|\mathbf{x})\right] \tag{13}$$

where $\mathbf{h}$ is expressed as a deterministic function of the form presented in Kingma & Welling (2013) and section 2.2:

$$\tilde{\mathbf{h}} = h(\epsilon, g, \tau, \phi) \tag{14}$$

with

$$\epsilon \sim \mathcal{N}(\mathbf{0},\mathbf{I}), \quad g \sim Gumbel(0,1)$$

We note that $q_\phi(\mathbf{h}|\mathbf{x})$ is used to denote the Concrete density and the KL term is computed according to eq. 20 in (Maddison et al., 2016). Finally, we also make use of batch normalization introduced by Ioffe & Szegedy (2015), since it has been shown to both improve convergence time and allow for the training of the upper layers in deep generative models Sønderby et al. (2016).

## 4 EXPERIMENTS

In our experiments we use two real world data sets of deidentified patient data, which correspond to flow cytometric measurements of two different blood samples [1] We investigate our model's ability to learn useful representations via its performance as an "unsupervised classifier". Detailed explanations of the data sets can be found in Appendix A.

### 4.1 METRICS AND EARLY STOPPING CRITERION

In our experiments we compare our model's predictive capacity with that of popular baselines from the field of computational FC, as well as generative models. Because in this particular problem one population (i.e. class) vastly outnumbers the other, one cannot rely on accuracy to correctly estimate the models' performance, as a model that would predict only the over-represented class

---

[1]Both data sets are publicly available with permission in `https://github.com/dimkal89/unsupervised_classification`

would consistently get high accuracy scores, but in reality would have limited predictive power. Instead, we make use of confusion matrix-based metrics, which are popular in medical applications. More specifically, we use *true positive rate* (TPR - also called *sensitivity* or *recall*) and *true negative rate* (TNR - also called *specificity*).

Because we found that training the model is difficult, as far as retaining optimal representations is concerned throughout the training procedure, we use early stopping, where we check the model's performance on the test set with a fixed frequency. To derive a useful criterion we also turn to a confusion matrix-based quantity, Matthews correlation coefficient (MCC), introduced by Matthews (1975). As a correlation coefficient, MCC takes values in [-1,1] with 1 suggesting that predictions and ground truth are in complete accordance, -1 that they are in complete discord and 0 that predictions are random. MCC is regarded as a good measure for estimating classifier performance on imbalanced data sets (Boughorbel et al., 2017). Formally, it is defined as follows:

$$MCC = \frac{TP \times TN - FP \times FN}{\sqrt{(TP + FP)(TP + FN)(TN + FP)(TN + FN)}} \tag{15}$$

We check MCC in the test set during training in fixed intervals and save the model parameters once it has crossed a set threshold. In general, we noticed in our experiments that a threshold of 0.5 yields good results with very small variance in performance.

## 4.2 RESULTS

In our experiments we investigate our model's predictive performance, i.e. its capacity to correctly classify cell measurements [2]. Our model's predictions are based on hard assignment of the probabilities obtained by the predictive distribution $q_\phi(\mathbf{y}|\mathbf{z})$. I.e. given a minibatch of observations $\mathbf{X} = \{\mathbf{x}^{(i)}\}_{i=1}^{N}$, probabilities of samples drawn from $q_\phi(\mathbf{y}|\mathbf{z})$ are thresholded at 0.5. Probabilities that lie above this threshold are considered positive cell measurements.

As baselines we choose the 3 best-performing algorithms from Weber & Robinson (2016) who compare a wide set of clustering algorithm implementations used in computational FC in two different tasks - multiple population identification and rare population detection. We are interested in the latter task which is closer to our own.

The first baseline, *X-shift* (Samusik et al., 2016) is based on k-Nearest Neighbors (kNN) density estimation. Local density maxima become centroids and the remaining points are connected to those centroids via ascending density paths. *Rclusterpp* (Linderman et al., 2013) is the second baseline and is an efficient implementation (with respect to memory requirements) of hierarchical clustering. The last baseline, *flowMEANS* (Aghaeepour et al., 2011) is based on k-means clustering but can identify concave cell populations using multiple clusters.

We argue that to successfully learn and separate the two manifolds of healthy and pathological cells, more expressive and distributed representations are necessary, with explicit steps taken during training to enforce these characteristics. Additionally, the proposed model must be a mixture of continuous and discrete distributions to appropriately represent different cell attributes and category respectively. To illustrate the merit in these points, we also include 2 methods from the VAE literature, a vanilla VAE (Kingma & Welling, 2013), followed by a linear support vector machine (denoted by VAE+SVM in tables 1 and 2) and a version of our own model but within the $\beta$-VAE framework (Higgins et al., 2016). The motivation for VAE+SVM is the fact that given adequate manifold separation, a linear classifier such as a support vector machine should, in principle, be able to correctly classify observations. The $\beta$-VAE framework introduces a hyperparameter $\beta$ to the variational objective, similar to $\lambda$, which we are using for deterministic warm-up. For $\beta > 1$, the model is forced to learn more compressed and disentangled latent representations, which is what Higgins et al. (2016) argue for. Learning disentangled representations is certainly closely related to successfully separating multiple data manifolds and we consider a representation to be efficient and disentangled in this scenario if it leads to good predictive performance, implying adequate manifold separation. In short we are treating "unsupervised classification" as a proxy for manifold separation. Finally to illustrate the merits of generative modeling in the task of separating widely overlapping

---

[2]Details on the experimental setup and the architectures used can be found in Appendix B

data manifolds we also include a deterministic 2-layer MLP classifier which is trained in a supervised way, i.e. using cross entropy as a cost function.

The results of our experiments can be shown in tables 1 and 2, where we denote our model by HCDVAE (Hierarchical Continuous-Discrete VAE).

Table 1: Patient data set 1

| Method | TPR | TNR |
|---|---|---|
| Rclusterpp | 0.5 | 0.935 |
| flowMEANS | 0.904 | 0.363 |
| X-shift | 0.434 | 0.901 |
| MLP | 0.0 | 1.0 |
| VAE+SVM | 0.043 | 1.0 |
| $\beta$-VAE | 1.0 | 0.878 |
| HCDVAE | **1.0** | **0.936** |

Table 2: Patient data set 2

| Method | TPR | TNR |
|---|---|---|
| Rclusterpp | 0.833 | 0.931 |
| flowMEANS | 0.264 | 0.999 |
| X-shift | 0.5 | 0.889 |
| MLP | 0.0 | 1.0 |
| VAE+SVM | 0.076 | 1.0 |
| $\beta$-VAE | 1.0 | 0.958 |
| HCDVAE | **1.0** | **0.994** |

The baselines developed for the domain of FC are in essence clustering algorithms and as such they suffer from issues clustering algorithms traditionally suffer from, the most important of which, are sensitivity to parameter configurations and dependence on initialization schemes. Concordantly, they exhibited high variance across multiple runs so we present the best results across 30 runs for each algorithm. Most of the baselines were able to achieve good predictive performance for the healthy cell population across most runs, which is to be expected since it is overrepresented in the data set, but average and erratic performance, for pathological cells. With the exception of Rclusterpp which achieved good overall performance in the second data set, all algorithms in both data sets exhibited a tendency to either be sensitive (high TPR) sacrificing healthy cell predictive accuracy or be specific (high TNR) and sacrifice pathological cell predictive accuracy.

The supervised MLP classifier is clearly unable to separate the two manifolds, "overfitting" the healthy cell population. This suggests that at least for binary problems in which classes are severely imbalanced, training supervised classifiers (i.e. training on log-likelihood objectives with explicit labeled information) results in subpar performance compared to generative modeling. This is not surprising as appropriate generative models incorporate more information about the data in their representations than just its labels, resulting in greater discriminative power. The performance of VAE+SVM suggests that the topmost latent layer of vanilla VAE is not able to separate the healthy/pathological manifolds. Presumably this is the case because the approximate posterior has collapsed early on the prior and rendered most units inactive, making the overall representation uninformative with respect to the global structure of the data. Deep VAE architectures are also known to be difficult to train, with top layers being slow to learn useful representations or even unable to learn at all. This is not the case, however with $\beta$-VAE and HCDVAE. The predictive performance of both approaches is similar, however opting for a more distributed representation seems to yield marginally better predictive capacity. Visually, the two approaches seem to result in similar latent representations as can be seen in Figures 3 and 4, where they collapse the majority of healthy cells into a compact cluster and separate the pathological cell manifold, remaining largely invariant to other factors present in the data. A possible explanation for the marginally better performance of HCDVAE is that its denser representation captures the activity of more explanatory factors in the data set, which $\beta$-VAE's representations miss due to excessive compression. We further hypoth-

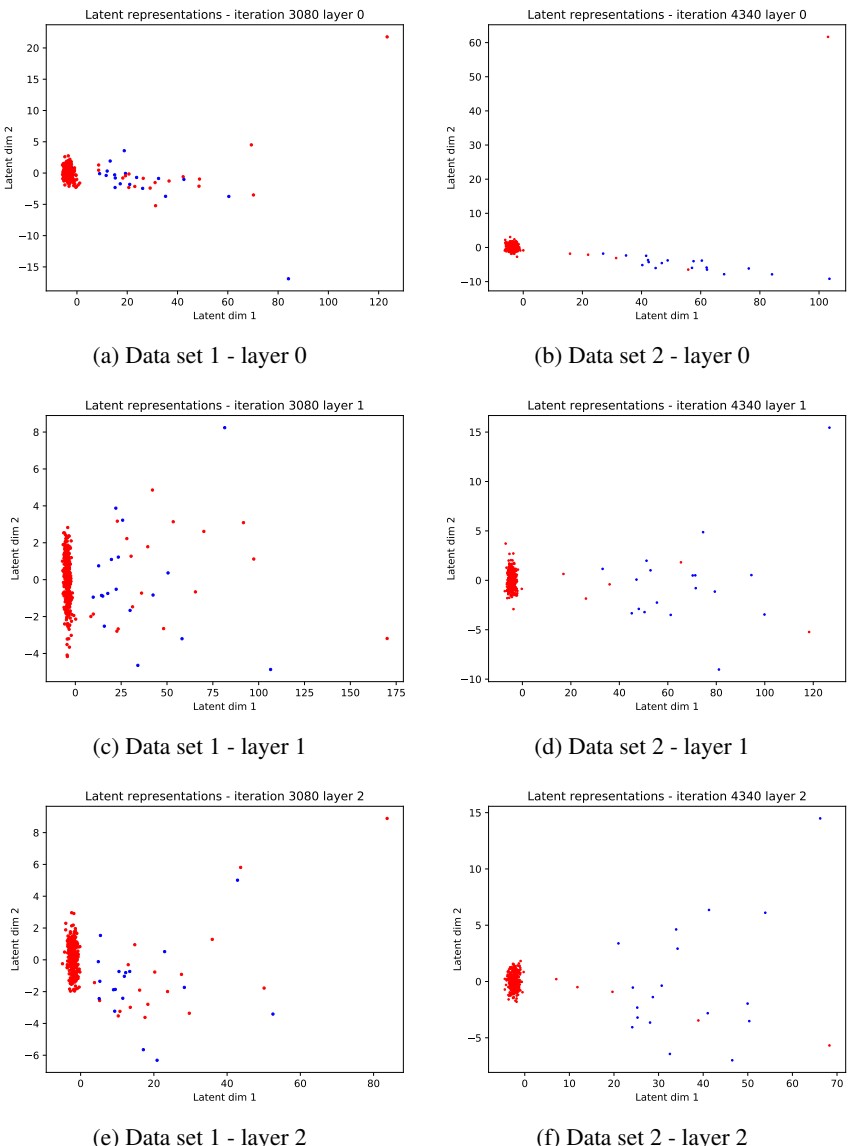

Figure 3: PCA plots of latent representations for experiments on both patient data sets for HCD-VAE. Red indicates healthy cells. Blue indicates pathological cells. For clarity only minibatches containing pathological cells are shown. Best viewed in color.

esize that the mixture of discrete and continuous variables, which HCDVAE and $\beta$-VAE share in our experiments provides a powerful posterior which benefits both approaches, resulting in similar performance. The discrete variable $\mathbf{y}$ at the top of the feature hierarchy representing cell state (or alternatively category) is largely invariant to perturbations in the factors modeled by the lower latent layers. Overall, HCDVAE achieves near human-level performance in both data sets, especially in the second case. We note that that the ground-truth diagnosis was crosschecked by 2 different human experts.

# 5 RELATED WORK

Models that use mixtures of discrete and continuous variables and are trained with the variational objective have previously been discussed in literature. Kingma et al. (2014) proposed a model for

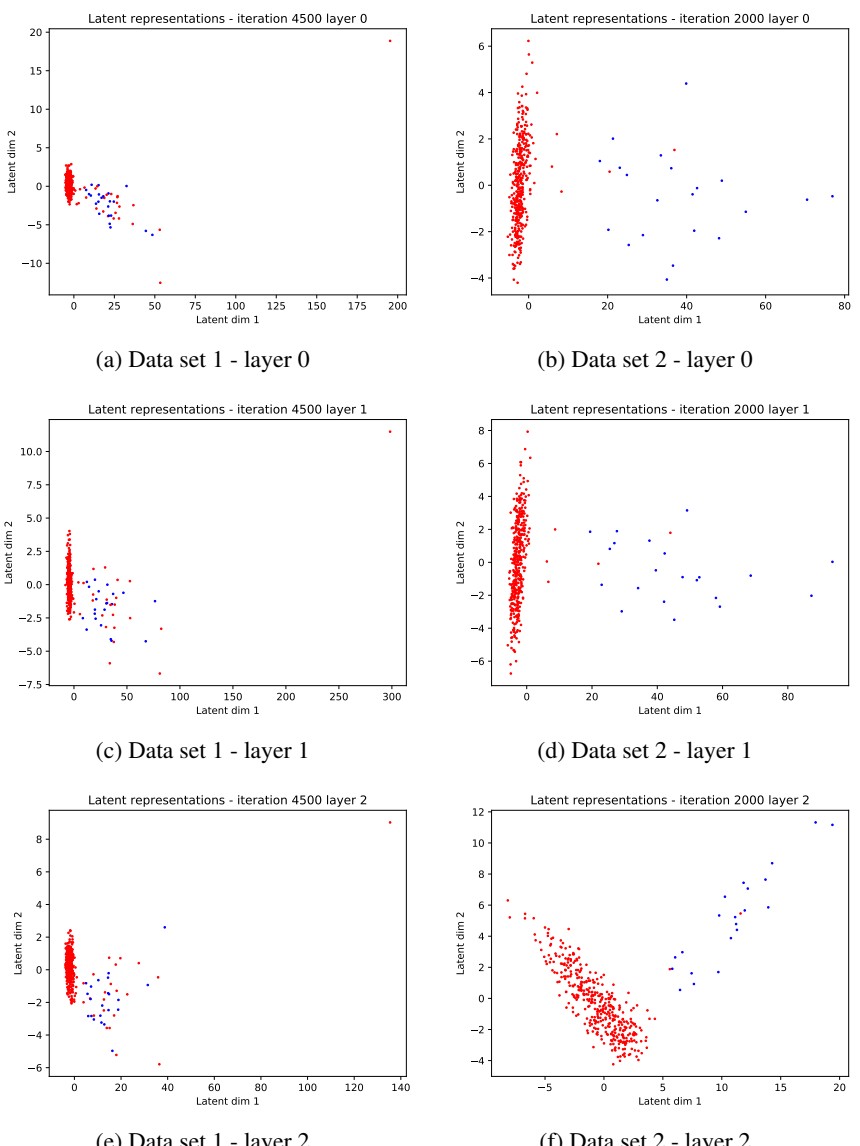

Figure 4: PCA plots of latent representations for experiments on both patient data sets for $\beta$-VAE. Red indicates healthy cells. Blue indicates pathological cells. For clarity only minibatches containing pathological cells are shown. Best viewed in color.

semi-supervised classification (M2) based on a discrete class variable $\mathbf{y}$ and a continuous "style" variable $\mathbf{z}$. Optimization is performed using the reparameterization trick for $\mathbf{z}$ and marginalizing over $\mathbf{y}$. This model is similar in structure to our own and Kingma et al. (2014) also raise the point of learning better discriminative features which are more clearly separable, making classification easier (M1 model). Another similarity is that the distribution $q_\phi(\mathbf{y}|\mathbf{x})$ can be used as a classifier, performing classification through inference. A crucial difference is that we are using a relaxed discrete variable, removing the need to marginalize over its possible values. Additionally, we are enforcing dependence between the continuous and discrete variables. Maaløe et al. (2017) presented a method that also employs discrete and continuous variables also in the context of semi-supervised learning where the continuous variables model the formation of natural clusters in the data and the discrete variables representing class information refine this clustering scheme. Johnson et al. (2016) use both discrete and continuous latent variables to construct a structured VAE that uses conjugate priors to create more flexible approximating posteriors in the context of switching linear dynam-

ical systems. Works that attempt to alleviate the non-informativeness of the prior have also been presented in literature. Dilokthanakul et al. (2016) present a VAE variant for the task of unsupervised clustering, introducing a Gaussian Mixture prior to enforce multi-modality on the inference distribution while employing the minimum information constraint of Kingma et al. (2016) to avoid cluster degeneracy. Goyal et al. (2017) develop tree-based priors, which they integrate into VAEs and train them jointly for learning representations in videos with the overall model being able to learn a hierarchical representation of the data set. Finally, Rolfe (2016) presented a model which combines a discrete component which consists of a bipartite Boltzmann machine with binary units on the one hand and a continuous component consisting of multiple continuous layers on the other. To perform optimization using the reparameterization trick, the binary variables are marginalized out. While the proposed "discrete VAE" learns the class of objects in images, it is significantly more complex in its architecture and still relies on marginalization of the discrete variables.

## 6 CONCLUSION

We have presented a framework for unsupervised feature learning and classification based on modifications of the original VAE framework (Kingma & Welling, 2013), making use of a mixture of relaxed discrete and continuous distributions, as well as batch normalization and deterministic warm-up (Sønderby et al., 2016), which we have found to be crucial for our framework's success. We trained our framework in a completely unsupervised manner, however relied on a stopping criterion that depends on labeled data. We have presented state-of-the-art results in a difficult real world problem having tested our approach against state-of-the-art computational FC and VAE models. Future work includes deriving a completely unsupervised early stopping criterion, adaptation to more problem domains and further principled investigation of the discrete variable to control the framework's modeling capacity.

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

## APPENDIX

## A    DATA SETS

Both data sets represent measurements on blood samples taken from patients and comprise 500,000 cells. Each cell measurement is represented by a 12-dimensional vector, corresponding to 10 different markers for each cell, the time stamp of the measurement and a label indicating healthy/pathological state. The first dimension corresponds to the time stamp, while the next 2 correspond to the forward and side scatter of the light. The remaining dimensions correspond to the protein markers used to make a diagnosis. The data sets' last column represents the label for each measurement. The data set used in the first experiment contains 107 pathological cells, while the one used in the second experiment contains 103 pathological cells. All patient identifiers are removed and the data sets are anonymized.

## B    DETAILED EXPERIMENTAL SETUP

For our experiments we shuffle and then split the data sets to training and test sets with a 75/25 split. To speed up training, we ensure that there is at least one example of a pathological cell in every minibatch and reshuffle the training set every 20 iterations. We used early stopping to stabilize training, with the stopping criterion being a threshold score of 0.5 for MCC, as reported in the main text.

For these experiments we used 3 layers of latent variables of 128, 64 and 32 units respectively. The discrete variable was chosen to be a Bernoulli variable and $p_\theta(\mathbf{x}|\mathbf{h})$ is a Gaussian. In the inference model, every Gaussian latent layer is parameterized by feedforward neural networks with 2 hidden layers of 256 units each. The non-linearity used is the rectifier function, while the mean and variance parameters are given by linear and softplus layers respectively. The mean parameter

of the relaxed Bernoulli distribution is also given by a feedforward neural network with the same architecture as above. For the generative model, every Gaussian latent layer is also parameterized by feedforward neural networks with 2 hidden layers with 128 rectifier units each. The mean and variance parameters are computed as before. All networks are optimized with minibatch gradient descent using the Adam optimizer (Kingma & Ba, 2014) with an initial learning rate of $10^{-4}$ which we decay exponentially for 3500 steps with a basis of 0.99. The minibatch size was set at 100. The relaxation parameter $\tau$ of the relaxed Bernoulli distribution was fixed at 0.66 per Maddison et al. (2016). The $\lambda$ term was linearly annealed from 0 to 1 with a step of 0.002. For the baselines, all parameter values were chosen according to Weber & Robinson (2016). The above architecture is shared between the model we denote by $\beta$-VAE and HCDVAE in tables 1 and 2. The VAE part of the model we denote by VAE+SVM has the same Gaussian layer architecture.

