# OpenReview forum: "Towards Unsupervised Classification with Deep Generative Models"
_ICLR.cc/2018/Conference — Reject_

### Official Review · AnonReviewer1 · 2017-11-28
**Apparently impressive result, but very little novelty**

**Rating:** 4
**Confidence:** 4

**Review:**

This paper addresses the question of unsupervised clustering with high classification performance. They propose a deep variational autoencoder architecture with categorical latent variables at the deepest layer and propose to train it with modifications of the standard variational approach with reparameterization gradients. The model is tested on a medical imagining dataset where the task is to distinguish healthy from pathological lymphocytes from blood samples.

I am not an expert on this particular dataset, but to my eye the results look impressive. They show high sensitivity and high specificity. This paper may be an important contribution to the medical imaging community.

My primary concern with the paper is the lack of novelty and relatively little in the way of contributions to the ICLR community. The proposed model is a simple variant on the standard VAE models (see for example the Ladder VAE https://arxiv.org/abs/1602.02282 for deep models with multiple stochastic layers). This would be OK if a thorough evaluation on at least two other datasets showed similar improvements as the lymphocytes dataset. As it stands, it is difficulty for me to assess the value of this model.

Minor questions / concerns:

- The authors claim in the first paragraph of 3.2 that deterministic mappings lack expressiveness. Would be great to see the paper take this claim seriously and investigate it.
- In equation (13) it isn't clear whether you use q_phi to be the discrete mass or the concrete density. The distinction is discussed in https://arxiv.org/abs/1611.00712
- Would be nice to report the MCC in experimental results.

---

### Official Review · AnonReviewer3 · 2017-11-29
**TOWARDS UNSUPERVISED CLASSIFICATION WITH DEEP GENERATIVE MODELS**

**Rating:** 4
**Confidence:** 4

**Review:**

The authors propose a deep hierarchical model for unsupervised classification by using a combination of latent continuous and discrete distributions.

Although, the detailed description of flow cytometry and chronic lymphocytic leukemia are appreciated, they are probably out of the scope of the paper or not relevant for the presented approach.

The authors claim that existing approaches for clustering cell populations in flow cytometry data are sensitive to noise and rely on cumbersome hyperparameter specifications, which in some sense is true, however, that does not mean that the proposed approach is less sensitive to noise or that that the proposed model has less free-parameters to tune (layers, hidden units, regularization, step size, link function, etc.). In fact, it is not clear how the authors would be able to define a model architecture without label information, what would be the model selection metric to optimize, ELBO?. At least this very issue is not addressed in the manuscript.

In Figure 1, please use different colors for different cell types. It is not described, but it would be good to stress out that each of the 4 components in Figure 1 right, corresponds to a mixture component.

The results in Tables 1 and 2 are not very convincing without clarity on the selection of the thresholds for each of the models. It would be better to report threshold-free metrics such as area under the ROC or PR curve. From Figures 3 and 4 for example, it is difficult to grasp the performance gap between the proposed approach and \beta-VAE.

- FC and CLL are not spelled out in the introduction.
- Equation (5) is confusing, what is h, y = h or is h a mixture of Gaussians with \alpha mixing proportions?
- Equation (6) should be q(z_L|z)
- Equation (8) is again confusing.
- Equation (10) is not correct, x can't be conditioned on h, as it is clearly conditioned on z_1.
- Equation (11) it should be q_\phi().
- It is not clear why the probabilities are thresholded at 0.5
- Figures 3 and 4 could use larger markers and font sizes.

---

### Official Review · AnonReviewer4 · 2017-12-07
**Interesting results, weak novelty, unjustified model choices.**

**Rating:** 4
**Confidence:** 5

**Review:**

Summary

The authors propose a hierarchical generative model with both continuous and discrete latent variables. The authors empirically demonstrate that the latent space of their model separates well healthy vs pathological cells in a dataset for Chronic lymphocytic leukemia (CLL) diagnostics.


Main

Overall the paper is reasonably well written. There are a few clarity issues detailed below.
The results seem very promising as the model clearly separates the two types of cells. But more baseline experiments are needed to assess the robustness of the results.

Novelty

The model introduced is a variant of a deep latent Gaussian model, where the top-most layer is a discrete random variable. Furthermore, the authors employ the Gumbel-trick to avoid having to explicitly marginalize the discrete latent variables.

Given the extensive literature on combining discrete and continuous latent variables in VAEs, the novelty factor of the proposed model is quite weak.

The authors use the Gumbel-trick in order to avoid explicit marginalization over the discrete variables. However, the number of categories in their problem is small (n=2), so the computational overhead of an explicit marginalization would be negligible. The result would be equivalent to replacing the top of the model p(y) p(z_L|y) by a GMM p_{GMM}(z_L) with two Gaussian components only.
Give these observations, it seems that this is an unnecessary complication added to the model as an effort to increase novelty.
It would be very informative to compare both approaches.

I would perhaps recommend this paper for an applied workshop, but not for publication in a main conference.

Details:

1) Variable h was not defined before it appeared in Eq. (5). From the text/equations we can deduce h = (y, z_1, …, z_L), but this should be more clearly stated.
2) It is counter-intuitive to define the inference model before having defined the generative model structure, perhaps the authors should consider changing the presentation order.
3) Was the VAE in VAE+SVM also trained with lambda-annealing?
4) How does a simple MLP classifier compares to the models on Table 1 and 2?
5) It seems that, what is called beta-VAE here is the same model HCDVAE but trained with a lambda that anneals to a value different than one (the value of beta). In this case what is the value it terminates? How was that value chosen?
6) The authors used 3 stochastic layers, how was that decided? Is there a substantial difference in performance compared to 1 and 2 stochastic layers?
7) How do the different models behave in terms train vs test set likelihoods. Was there overfitting detected for some settings? How does the choice of the MCC threshold affects train/test likelihoods?
8) Have the authors compared explicit marginalizing y with using the Gumbel-trick?

Other related work:

A few other papers that have explored discrete latent variables as a way to build more structured VAEs are worth mentioning/referring to:

[1] Dilokthanakul N, Mediano PA, Garnelo M, Lee MC, Salimbeni H, Arulkumaran K, Shanahan M. Deep unsupervised clustering with gaussian mixture variational autoencoders. arXiv preprint arXiv:1611.02648. 2016 Nov 8.

[2] Goyal P, Hu Z, Liang X, Wang C, Xing E. Nonparametric Variational Auto-encoders for Hierarchical Representation Learning. arXiv preprint arXiv:1703.07027. 2017 Mar 21.

---

### Author Response · Authors · 2018-01-04
**Response to reviewers**

We thank the reviewers for their feedback. We chose to respond with a top-level comment as some concerns were shared by the reviewers.

Regarding the novelty of the paper we felt that we achieved good results in two difficult real world data sets which relate to an important real world problem. We tried to present deep generative modeling as a viable solution to a scenario that is all too frequent in most real world settings (i.e. significantly imbalanced data sets).

During our experiments we noticed that explicit marginalization over the discrete variable was not in fact able to separate the two manifolds of interest. In fact the model "overfitted" the predominant class in the data set, completely ignoring the discrete latent variable. This phenomenon is consistent with previous analysis (please see https://openreview.net/forum?id=rydQ6CEKl and http://ruishu.io/2016/12/25/gmvae/). Thus, introducing the Gumbel-Softmax trick was not at all an effort to introduce artificial novelty but a choice made out of necessity. We agree that a comparison between the two approaches would be illuminating it is not entirely clear why the relaxed continuous density yields better predictive performance than the discrete mass, which seems to remain completely uninformative throughout training. We are still running experiments and considering information theoretic tools to analyze this phenomenon.

The number of stochastic layers was chosen with the nature of the task at hand in mind, i.e. we tried to maximize predictive performance, and so while 1 stochastic layer of 128 units was enough to achieve around 0.88 for TPR and 0.91 for TNR for the first data set and 0.9 for TPR and 0.93 for TNR, we increased the number of layers to optimal results. A greater number of layers, while still trainable did not yield better performance. We would also like to note that because of the above our model is much less sensitive to noise, as similar configurations of hyperparameters yielded similar results, which is not the case for the clustering baselines we compared it against.

We included an MLP classifier in the revision of our paper to better highlight the merit of generative modeling and stochastic features against deterministic mappings induced by training with labeled data in scenarios of significant class imbalance. As expected the classifier "overfitted" the predominant class in the data set.

The MCC early stopping criterion puts a limit on generative performance. I.e. if the model is trained for more iterations it can reach better log likelihood scores, however we note that according to our experiments, higher log-likelihood scores do not imply good predictive performance, which is also a reason we omitted them from the paper, since we focus on classification, rather than generative performance. Having said that, the model was not found to overfit the data either in terms of predictive or generative performance.

beta-VAE proposes the fixing of a \beta term to a constant value that is greater than 1, in an effort to encourage the learning of more efficient latent codes. I.e. it is not annealed. In our experiments we tried among {5, 25, 50, 100, 250, 500}, however values greater than 5 yielded no improvement in discriminative performance.

The threshold was chosen to be at 0.5 since we make the assumption that the Bernoulli probability represents the positive case (i.e. 1). This can be easily understood if one thinks about it in terms of a categorical variable with 2 components, each representing a diagnosis outcome (i.e. p(K=k1) = \theta and p(K=k2) = 1 - \theta).

---

### Decision · Program_Chairs · 2018-01-29
**ICLR 2018 Conference Acceptance Decision**

**Decision:**

Reject

**Comment:**

The authors propose a hierarchical VAE model with a discrete latent variable in the top-most layer for unsupervised learning of discriminative representations.  While the reported results on the two flow cytometry datasets are encouraging, they are insufficient to draw strong conclusions about the general effectiveness of the proposed architecture. Also, as two of the reviewers stated the proposed model is very similar to several VAE models in the literature. This paper seems better suited for a more applied venue than ICLR.